# Adipose Tissue Mitochondrial Factors Profile after Dietary Bioactive Compound Weight Reduction Treatments in a Mice Obesity Model

**DOI:** 10.3390/ijms20235870

**Published:** 2019-11-22

**Authors:** Anna Cassanye, Meritxell Martín-Gari, Manuel Portero-Otin, José CE Serrano

**Affiliations:** Department of Experimental Medicine, NUTREN-Nutrigenomics, University of Lleida, 25198 Lleida, Spain; annakasa4@gmail.com (A.C.); meritxell.martin@udl.cat (M.M.-G.); manuel.portero@mex.udl.cat (M.P.-O.)

**Keywords:** adipose tissue, fish oil, soluble fibre, soy, mitochondria, metabolism

## Abstract

Prolonged caloric intake above energy needs disturbs the body’s ability to store and manage the excess of energy intake, leading to the onset of chronic degenerative diseases. This study aimed to compare the effect of three foods, which contain demonstrated bioactive compounds in the treatment of obesity and as an adjuvant in obesity energy restriction treatments. In a mice obesity model induced through a high-fat diet; fish oil, soluble fibre, and soy were incorporated to evaluate its capacity to modulate metabolic factors in adipose tissue during a continued fat intake or weight reduction through a normocaloric diet. As a result, fish oil improved mitochondrial related, adipose tissue hormone expression, and oxidation products when high-fat diets are consumed; while soluble fibre improved glucose and inflammation pathways during high-fat diet intake. In weight reduction treatments few differential features, as a treatment adjuvant, were observed for fish oil and soy; while soluble fibre was able to improve the weight reduction effects induced by a normocaloric diet. As a conclusion, soluble fibre supplementation compared to an energy reduction program, was the only treatment able to induce a significant additional effect in the improvement of weight loss and adipose tissue metabolism.

## 1. Introduction

Severe obesity is an increasingly prevalent condition and is often associated with long-term comorbidities, reduced survival, and higher healthcare costs. Although several resources have been invested to treat and prevent obesity, an overall median success rate of 15% is observed in most studies and intervention programs [1,2]. While pharmacological strategies are indicated for patients with high cardiovascular risk, non-pharmacological treatments such as lifestyle interventions (diet, physical exercise, stress management, and others) are indicated for a great majority of people with overweight problems. This fact raises the interest of the health and scientific community to address additional tools to act as adjuvants in the treatment of obesity, some of them based on food technological modifications to induce satiety, while others through the addition of food bioactive compounds that may improve insulin signaling pathways, the regulation of adiposity, energy expenditure, and/or energy bioavailability [3].

Several studies indicate that fish oil and omega-3 polyunsaturated fatty acids supplements may help to prevent cardiovascular and metabolic diseases due to their anti-inflammatory properties. However, meta-analysis of clinical controlled trials does not show any improvement in body weight and lean body mass [4], while others have observed an improvement in blood lipid and metabolic parameters without significant changes in anthropometric parameters [5]. On the other hand, soy has been widely studied due to the quality of its protein and high isoflavone content. Lipogenesis inhibition and increased fatty acid β-oxidation, which lead to the reduction of body fat depots, are mechanisms of isoflavones action against obesity [6]. In this regard, significant effects in anthropometric parameters have been observed in low doses and during a short period of supplementation (two to six months); but adverse effects have been observed in overweight and obesity populations [7]. Lastly, fibre, especially soluble fibre, is widely recommended as an anti-obesity agent due to its ability to interfere with gut lipid absorption, satiety and satiation effect, and gut microbiota composition and activity modulation through short-chain fatty acids production between others [8].

Notwithstanding the above, the main problem during the development of obesity and its further co-morbidities is the body’s inability to manage energy overload, generally known as lipotoxicity [9]. Mitochondrial dysfunction in adipocytes possibly induced by lipotoxicity results in detrimental effects on its differentiation, lipid metabolism, insulin sensitivity, and oxidative capacity between others and has been reported to have a strong correlation with metabolic diseases, including obesity, type 2 diabetes and other metabolic diseases [10]. In this context, adipose tissue “healthy” metabolism is a key feature for the success of weight reduction in patients with morbid obesity after bariatric surgery [11]. Nonetheless, the main problem in this context is how to improve the metabolic capacity of adipose tissue of people with obesity that shows an unbalanced metabolism. Therefore, the main objective of this study was to determine the possible adjuvant effect in the adipose tissue metabolic profile during weight reduction-treatments of three recognized dietary treatments: Fish oil, soluble fibre, and soy; in adipose tissue metabolism after the onset of obesity. A mice obesity model was employed where transcriptomic and oxidative markers analyses were employed to determine changes in adipose tissue metabolism induced by the selected dietary treatments.

## 2. Results

The study was conducted in two periods. In the first period, there was an induction of obesity through a high-fat diet for four months with an observed mean increase in body weight of about 10 g (58.8 ± 3.9 versus 48.3 ± 3.6 in animals fed with a high-fat diet and normocaloric diet, respectively) (Figure 1A). Once overweight was induced, the treatment period started in which, different bioactive compounds as adjuvants in weight reduction treatments were added to normocaloric and high-fat diets.

For a better understanding of the results from the present study, the behavior observed in the overweight animals that were subjected to a normocaloric diet for two months (in order to reduce weight) and the two controls of the study: (1) Animals that were always fed with a normocaloric diet; and (2) overweight animals that continued during two months with a high-fat diet; will be described firstly. As shown in Figure 1A, the gap in the weight of the animals fed during six months with a high-fat and normocaloric diet continued increasing in this time period, with a mean difference between the two groups of 12.5 g (64.6 ± 4.3 versus 52.1 ± 5.2, respectively). The treatment with a normocaloric diet induced an accelerated weight reduction that was appreciable until week three of treatment; later slight weight reductions were observed with a final difference between normocaloric control group and normocaloric weight reduction group of 2.4 g. Concerning dietary intake (Table 1), it was observed that a higher intake of the normocaloric diet, due to a lower energy density; compared to the high-fat diet (0.55 versus 0.47 g/week/g body weight, respectively). However, from the energy point of view, the amount of energy ingested was lower in the weight reduction group compared to the two controls (2.1 ± 0.2, 2.7 ± 0.5, and 2.8 ± 0.7 kcal/week/g, respectively).

The weight reduction improved the capacity for homeostatic regulation of glucose as seen in Figure 1B,C; where similar behaviors in the changes in blood glucose in the normocaloric control group and normocaloric weight reduction group were observed in both tolerance tests. Despite the differences in the capacity to regulate glycaemia, no differences in blood biochemical parameters were observed between the three groups (Table 1), although the normocaloric treated group presented a healthier lipid profile.

To determine the effects on energy availability in different metabolic parameters in the adipose tissue, a transcriptomic and protein oxidative damage analysis was performed. At the transcriptional level, it is observed that the high-fat diet induces an increase in the levels of cytochrome b-245 alpha (*Cyba*) (Figure 1D), a subunit of the enzyme complex NADPH oxidase, related to the generation of oxidative stress. Despite this change, no differences were observed in protein oxidative stress in the adipose tissue between normocaloric and high-fat diet controls (Table 2). Notwithstanding, after the weight reduction through a normocaloric diet, a reduction in malondialdehyde-lysine (MDAL) levels, a marker of oxidative protein damage from fat peroxidation, was observed. Regarding the other transcriptional parameters analyzed, no significant differences were observed between normocaloric, high-fat, and normocaloric treated diet group (Appendix A). To determine if the transcriptional and protein oxidative stress parameters could discriminate metabolic differences in adipose tissue between the three groups, a partial least square-discriminant analysis (PLS-DA) analysis was performed (Figure 1E,F). It is observed that the differences in selected RNA and oxidative stress levels can differentiate between the three experimental groups with a probability of 61% (random forest OOB = 0.389), being high levels of *Tnfa, Nrf1, Tfam*, CML, and *Pparg* the ones that best characterize overweight animals that were subsequently treated with a normocaloric diet.

The differences in energy intake and availability induced changes in the number of mitochondria in the adipose tissue. Compared with the control group, a higher level of mtDNA was observed in the mice fed with a high-fat diet, while similar amounts were observed with the normocaloric diet treated group (Figure 2A). Despite the increase in mtDNA, the ratio porin/mtDNA was lower in the mice fed with a high-fat diet; while mice treated with a normocaloric diet showed increased ratios of porin/mtDNA (Figure 2B).

Concerning the effect of the bioactive compound (soy, fibre, and fish oil) supplementation during a high-fat feeding in overweight mice; it was observed that none of the bioactive foods had a significant effect in the prevention of the increase in body weight gain induced by the high-fat diet (Figure 3A). Similarly, no significant differences were observed in blood biochemistry parameters, except for a reduction in plasma antioxidant capacity observed in the soluble fibre treated group (Table 1). Regarding glucose homeostatic behavior, soluble fibre incorporation to a high-fat diet, induced a better capacity to regulate blood glucose levels (Figure 3B,C) in both glucose and insulin tolerance tests; while the treatments with fish oil and soy did not show significant differences. At a transcriptional level, it was observed that fish oil treatment induced higher levels of *Cyba* (Figure 3D), *Fasn,* and *Lep* (Appendix A) compared to the high-fat diet control group. About oxidative damage, soluble fibre and soy treatments increased SAG and decreased MDAL levels; while fish oil induced a decrease in both markers of oxidative protein modifications. In addition to the slight difference between the three groups in the levels of almost all analyzed parameters, the PLS-DA analysis showed that it is possible to discriminate the belonging of the adipose tissue within the three treatments groups with a probability of 94% (random forrest OOB = 0.0556), especially between fish oil and soluble fibre-soy treated groups (Figure 3E). The main variables that explain these differences are the higher levels of *Lep*, *Fasn*, *Tfam, Adipoq, Cyba,* and *Nrf1* and lower levels of SAG and CML observed in the fish oil treated group compared with soy and soluble fibre (Figure 3F).

Bioactive foods induced changes in the levels of mtDNA mainly a lower amount in the fish oil, a higher amount in soluble fibre, and no modifications in soy groups. Regarding the porin/mtDNA ratio, higher ratios were observed in the fish oil and soy group, while soluble fibre induced a lower porin/mtDNA ratio (Figure 2).

Finally, the adjuvant effect of fish oil, soluble fibre and soy incorporated in normocaloric diet to induce weight reduction did not show differences in the magnitude of weight reduction compared with the normocaloric diet alone; although the soluble fibre normocaloric diet was the treatment that induced the highest weight reduction between the four groups (Figure 4A). Notwithstanding, appreciable differences were observed in blood biochemistry parameters where fish oil and soluble fibre induced higher reductions in total cholesterol, and additionally a reduction in LDL-cholesterol levels by fish oil. The incorporation of soluble fiber into the normocaloric diet provides additional improvement in glycemic homeostasis, observed both from the glucose and insulin tolerance tests (Figure 4B,C). In relation to protein oxidative damage, it was observed that the incorporation of soluble fiber and soy induced an increase in SAG, and decrease in MDAL levels by soy (Table 2); while the levels of *Fasn, Pparg, Adipoq,* and *Dgat1* were significantly increased with the addition of soluble fiber (Appendix A). Unlike the previous observations, none of the adjuvant treatments induced a difference in the expression of *Cyba* (Figure 4D). Finally, transcription factors and the markers of protein oxidation were able to discriminate adipose tissues of the animals subjected to the different treatments with a probability of 83% (random forest OOB = 0.167), mainly between fish oil and soluble fibre-soy treated groups. Among the parameters that allow to discriminate between the three groups, the treatments with soy induced higher levels of SAG, CML, and 2SC, while soluble fibre induced higher levels of *Irs1* and *Dgat1*; and fish oil with higher levels of MDAL (Figure 4E,F).

Regarding the induced changes in mitochondria, when the selected dietary compounds were incorporated in the normocaloric diet, an increase in the levels of mtDNA and a reduction in the levels of porin/mtDNA ratio was observed in all groups (Figure 2).

Due to the complexity of the results, a brief review of the main effects of each treatment in different pathways in adipose tissue metabolism is described in Figure 5. For instance, fish oil seems to be a good candidate to improve mitochondrial related and adipose tissue hormone expression; and a reduction in oxidation products when high-fat diets are consumed. While soluble fibre may improve glucose and inflammation pathways during high-fat diet intake. In weight reduction treatments by energy adequacy through a normocaloric diet few differential features as a treatment adjuvant were observed for fish oil, soy, or soluble fibre. Soy incorporation may increase transcriptional factors related to mitochondria, while fish oil and soluble fibre may reduce parameters related to oxidation and inflammation, respectively.

## 3. Discussion

The objective of this study was to compare the effectiveness of different bioactive foods in the ability to reduce body weight when obesity and metabolic impairments are established; and the effect they would have on adipose tissue metabolism. Three foods that are a source of bioactive compounds were selected, such as soy (source of isoflavones, protein fraction, among others), fish oil (source of omega-3 fatty acids), and soluble fibre from Psyllium platago ovata. The three sources of bioactive compounds have been described in both animal models and humans, as foods capable of preventing and treating insulin resistance in obesity situations. However, the mechanisms of action described are different for each one of them, for example, omega 3 fatty acids are anti-inflammatory agents; soluble fibre through physical mechanism influencing satiety and energy absorption of food, as well as physiological through colonic fermentation by-products; while soy could improve insulin secretion and sensitivity [12]. In general terms, it was observed in this study that except soluble fibre, neither soy nor fish oil were able to reverse the adverse effects of a high-fat diet, nor did they show an adjuvant effect in normocaloric dietetic treatment. Although logically, it is observed that the most effective treatment for situations of obesity and insulin resistance was the reduction of energy intake.

The reduction of energy intake, by itself, tends to induce metabolic changes in adipose tissue related to mitochondria factors, based on the PLS-DA analysis such as higher levels of *Nrf1, Tfam,* and *Pparg*. However, the observed effect, no changes were observed in the levels of mtDNA between the normocaloric controls and the obese mice subjected to weight reduction through a normocaloric diet. Likewise, an increase in mtDNA levels was observed in obese mice fed with a high-fat diet. Suggesting that the similarity in the content in mtDNA in the normocaloric treated mice compared to the normocaloric control may represent a healthy feature. However, the increase in the ratio of porin/mtDNA observed in the normocaloric treated group suggests that this group may have a higher metabolic rate compared to normocaloric and high-fat controls. Since porin (also called voltage-dependent anion channel, VDAC) regulates the transport of all metabolites that enter and leave the mitochondria (except for a relatively few membrane-permeant lipophilic compounds such as oxygen, acetaldehyde, short-chain fatty acids); the increase of its content may suggest an increase in mitochondrial metabolic capacity [13]. Similar results have been reported in humans on the effect of weight reduction in adipose tissue, where an increase in the expression of genes related to oxidative phosphorylation, tricarboxylic acid cycle, and fatty acid beta-oxidation has been observed [14]. This greater capacity for energy metabolism is reflected in the energy intake of normocaloric treated overweight animals, where they consumed on average fewer calories per g of weight (2.13 kcal/g body weight) than mice with high-fat diets (2.43 kcal/g body weight) and control mice exposed throughout the treatment period to a normocaloric diet (2.67 kcal/g body weight). Suggesting in this way, the use of energy reserves of adipose tissue as a compensatory source to the reduced energy obtained by the diet. As shown in Table 1, the addition of fish oil and soy to normocaloric diets did not induce significant changes in energy intake (values between 2.05 to 2.13 kcal/g of body weight, respectively) nor body weight; while soluble fibre reported the lowest energy intake, highest weight reduction, and better homeostatic capacity of glucose metabolism. This suggests that the regulation of energy availability may be a key factor for obesity and its metabolic imbalance treatment. It is necessary to mention that the exposure to high-fat diets for four months was enough to induce metabolic inflexibility conditions; where the reduction of body weight may be impaired due to the disability of energy utilization [15].

In this sense, the next question is whether these bioactive foods may have a treatment effect in established obesity when high-energy intake is persistent. In this situation, the diet supplementation with soluble fibre seems to be the best treatment. It was observed that a better ability to regulate glucose homeostasis and to induce a slight reduction in body weight gain compared with other experimental groups. These effects are also described in humans. In a recent meta-analysis, a reduction of 0.84 points in the BMI, 0.41% in the percentage of body fat and 2 cm in the waist circumference is observed due to soluble fibre consumption [16]. Concerning the adipose tissue metabolism, this capacity to reduce body weight could be explained by an improvement in the capacity of glucose metabolism evidenced by higher levels of *Irs1*, CEL, and 2SC; and the greater sensitivity to glucose and insulin tolerance tests. This behavior has also been observed in humans, where patients with higher levels of CEL and 2SC in adipose tissue, have a greater capacity for weight reduction after bariatric surgery [11]. About the transcription factors related to mitochondria, based on the PLS-DA analysis soluble fibre treatment was characterized to present the lowest levels of *Tfam*. This observation may be contradictory since adipose tissue in obesity tends to be characterized by mitochondrial dysfunction [17]. Although TFAM is necessary for the stability of mtDNA and the initiation of transcription and replication of mtDNA, some authors suggest that its downregulation in situations of high-fat diets could be related to a greater mitochondrial oxidation capacity and higher oxygen consumption [18]. The mechanism of action by which soluble fibre induces an improvement when high-fat diets are taken, could be related either to the decrease in energy availability [19] or to the indirect effects of colonic fermentation products. In this sense, it has been observed that oral butyric acid can prevent the insulin resistance induced by high-fat diets, by an increase in the capacity of energy metabolization [20].

Fish oil, on the other hand, is the treatment with the most differential characteristics with respect to soluble fibre and soybean treatments (Figure 3E). It is characterized, based on the PLS-DA analysis by having lower levels of oxidative damage (MDAL, CML, SAG); and higher levels of transcription of *Lep* and *Adipoq*; inflammation (*Tnfa, Il6*); and genes related to the mitochondria (*Tfam, Nrf1,* and *Cyba*). Possibly the main effect of omega 3 fatty acids in the adipose tissue is to favor its expansion capacity, reflected by higher levels of *Fasn* and *Lep* in high-fat diets. This effect may be related to the anti-inflammatory capacity of the omega-3 fatty acids previously described in several studies [21,22], which induce higher levels of pro-resolving lipid mediators such as resolvins, protectins, and maresins. However, although it has a better transcriptional and oxidative damage profile, no significant differences are observed in the homeostatic capacity to regulate the glycemia observed in both glucose and insulin tolerance tests, or to prevent weight gain. Which suggests that the observed changes in mtDNA and porin/mtDNA ratio are not related to an improvement in energy homeostasis. Similar results have been observed in humans [23] and in mice where supplementation with eicosapentaenoic acid does not prevent weight gain in high-fat diets, or induce changes in oxygen consumption [24], and similar behaviors have been observed in other tissues such as skeletal muscle [25]. Finally, supplementation with soy did not show significant improvements compared to the high-fat diet. The potential discrepancy of these results with those previously published may be derived from the complexity in the nutritional composition of soybeans. For example, differences in the impact on body weight have been observed in treatments with soy and isoflavones [7]. Although isoflavones demonstrate anti-obesity properties, soy in obese subjects and when consumed in high quantities may induce an increase in body weight.

Some limitations of this study can be raised. Although there were observable differences in the transcriptomic profile, some of these changes in its levels may not be translated to real situations due to the regulation of protein expression or mitochondrial functionality. In this sense, different profiles of mtDNA were observed between treatments and no relationships were observed between the molecular changes and the homeostatic capacity to regulate glucose metabolism. Therefore, it is suggested for further studies to perform additional analyses of mitochondrial enzyme activity, protein expression, and/or respiratory capacity to determine the metabolic capacity of the adipose tissue. Moreover, the influence of the studied compounds in other organs and systems involved in energy metabolism such as liver and muscle are not addressed in this study. For example, it has been observed that normocaloric diet increase muscle *Glut4* mRNA levels in obese mice [26]. This is particularly important since not all organs and systems respond in a similar way to an increase or decrease in energy availability. Similarly, in the clinical setting, the effect of a mild reduction in body weight induces a quick switch from a disease to a clinically non-pathological condition. However, in this respect, frailty from this condition is observed if there is an increase in body weight. This implies that although some biomarkers are normalized during weight reduction programs, the metabolic system is unstable and liable to return to a disease condition. In other words, some organs may still have metabolic inflexibility, explaining in this way the deleterious effects of “yo-yo” dieting [27]. Notwithstanding, based on the information collected in this model, it is possible to conclude that besides soluble fibre, the diet supplementation of the selected bioactive foods, compared to an energy reduction program, does not induce a significant additional effect in the improvement of adipose tissue metabolism.

## 4. Materials and Methods

### 4.1. Animals and Diets

Three-week-old male CD1 Swiss (initial weight 22.1 ± 1.9) mice obtained from Harlan Laboratories (Catalunya, Spain) were maintained at 23 ± 2 °C under a 12:12 h light-dark cycle (lights on from 07:00 to 19:00). All mice were allowed unlimited access to in-house produced high-fat (Experimental and high-fat control group *n* = 48) or normocaloric diet (Control group *n* = 6) for four months. After that period, 24 animals that were fed with a high-fat diet were randomly assigned to a normocaloric diet (weight reduction control group *n* = 6) and a normocaloric diet + dietary bioactive compounds mainly fish oil (Sigma-Aldrich, F8020 Lot#051M1861V, St-Louis, MO, USA), soy (Vivesoy, Grupo Leche Pascual, Aranda del Duero, Spain), and soluble fibre from psyllium plantago (Laboratorios Normon S.A., Madrid, Spain) (*n* = 6 for each experimental group) for two months. The other half of the animals continued to be exposed to a high-fat diet (High-fat control group *n* = 6) and high-fat diet + dietary bioactive compounds mainly fish oil, soy, and soluble fibre (*n* = 6 for each experimental group) for two months. Sample size was estimated with an accepted risk of 0.05 and an estimated power of 90%. It was considered that the treatment would be effective if it induces a reduction of 6.5 g of body weight in the following two months of treatments; and a standard deviation of body weight of 3.6 g was used for sample calculation (estimated *n* = 5.25). The scheme of the experimental procedure is described in Appendix A, and the dietary and nutritional composition of experimental diets are described in Appendix A. Body weight, food and beverage intake were measured weekly throughout the study. At the end of the experimental period, subcutaneous glucose and insulin tolerance tests were performed (see below). Further, all animals were sacrificed and epididymal fat pad (visceral adipose tissue) was collected, frozen in liquid nitrogen, and stored at –80 °C. All animal procedures followed the approved protocols from the Institutional Animal Care and Use Committee and by the Ethics Committee of the University of Lleida (Approval number: CEEA 18-01/12; date: 2/2/2012).

### 4.2. Subcutaneous Glucose and Insulin Tolerance Tests

Mice were fasted for 12 or 2 h and then injected subcutaneously with glucose (2 g/kg body weight) or insulin (0.5 U/kg body weight, Sigma I0516 from bovine pancreas), respectively, with a rest period of five days between both tests. Blood samples were taken every 20 min (0–120 min) from the tail vein and blood glucose levels were determined with a portable glucometer (Roche Accu-Chek Aviva, CAT/TYP 05911982002, Mannheim, Germany). The area under the curve was calculated as the sum of trapezoids in both tests.

### 4.3. Adipose Tissue Homogenization

Samples were homogenized in a buffer containing 20 mM Tris pH:8, NaCl 150 mM, 2 mM EDTA, 1% Triton X-100, 10% Glycerol, 1 μM butylated hydroxyl toluene, 10 μg/mL aprotinin, 1 mM NaF, 1 mM Na_3_VO_4_, and a protease inhibitor mix (GE Healthcare 80-6501-23, Chicago, IL, USA) (1% *v/v*) with a Potter-Elvehjem device, at 4 °C. After samples were centrifuged (14,000 rpm, 20 min, 4 °C) the lower phase was collected for further analysis. Protein concentrations were measured using the Bradford protein assay (BioRad Laboratories, München, Germany).

### 4.4. Analysis of Protein Oxidative Modifications

Glutamic semialdehydes (GSA), glycoxidation (carboxyethyl-lysine (CEL) and carboxymethyl-lysine (CML)), lipoxidation (Malondialdehyde-lysine, MDAL) and 2-succinocystein (2SC) were determined by GC/MS as trifluoroacetic acid methyl esters derivatives in acid-hydrolyzed delipidated and reduced protein samples using a HP6890 Series II gas chromatograph (Agilent, Barcelona, Spain) with a MSD5973A Series and a 7683 Series automatic injector with a Rtx-5MS Restek column (30 m × 0.25 mm × 0.25 μm). Quantification was performed by internal and external standardization using standard curves constructed from mixtures of deuterated and non-deuterated standards. Analyses were carried out by selected ion monitoring GC/MS (SIM-GC/MS). The ions used were: Lysine and [^2^H_8_]lysine, m/z 180 and 187, respectively; 5-hydroxy-2-aminovaleric acid and [^2^H_5_]5-hydroxy-2-aminovaleric acid (stable derivatives of GSA), CML and [^2^H_4_]CML, m/z 392 and 396, respectively; CEL and [^2^H_4_]CEL, m/z 379 and 383, respectively, MDAL and d8-MDAL, m/z 474 and 482, respectively; and 2SC and d2-2SC, m/z 284 and 286. The amounts of products are expressed as the μmolar ratio of GSA, CML, CEL, MDAL, and 2SC per mol of lysine.

### 4.5. RNA Purification and Gene Expression Measurements

RNA was prepared using RNeasy Lipid Tissue Mini Kit (Qiagen, Izasa, Barcelona, Spain). The integrity of each RNA sample was checked by the Agilent Bioanalyzer (Agilent Technologies, Palo Alto, CA, USA). Total RNA was quantified by a spectrophotometer (GeneQuant; GE Health Care, Piscataway, NJ, USA) reverse transcribed to cDNA with the High Capacity cDNA Archive Kit (Thermo Fisher Scientific, Waltham, MA, USA) according to the manufacturer’s protocol. Gene expression was assessed by real-time PCR with a LightCycler 480 Real-Time PCR System (Roche Diagnostics, Barcelona, Spain), using TaqMan and SYBR green technology suitable for relative gene expression quantification. TaqMan probes for *Fasn, Pparg, Irs1, Adipoq, Lep, Plin1, Dgat1*, *Tfna, Il6, Cyba, Tfam, Nrf1,* and *18S* are detailed in Appendix A. The RT-PCR reaction was performed in a final volume of 12 μL. The cycle program consisted of an initial denaturing of 10 min at 95 °C then 40 cycles of 15 s denaturing phase at 95 °C and 1 min annealing and extension phase at 60 °C. Fold changes compared with the endogenous control were then determined by calculating 2^−∆Ct^. The gene expression results are expressed as the expression ratio relative to *18S* gene expression according to the manufacturer’s guidelines. No difference in the expression of *18S* was observed between groups (*p* = 0.1848).

### 4.6. Antioxidant Capacity

Antioxidant capacity was measured by the Ferric Reducing Antioxidant Power (FRAP) method. Briefly, 900 μL of the FRAP reagent, containing, 2,4,6-Tris(2-pyridyl)-s-triazine, FeCl_3_, and acetate buffer, was mixed with 90 μL of distilled water and 30 μL of the test sample or the blank (solvents used for homogenization). Maximum absorbance values taken at 595 nm were taken every 15 s at 37 °C. The readings at 30 min were selected for calculations of FRAP values. Solutions of known Trolox (Sigma-Aldrich, 238813, USA) concentration were used for antioxidant capacity equivalents.

### 4.7. Blood Biochemical Analysis

Blood biochemical profile was determined by an enzymatic spectrophotometric test from Spinreact (St. Esteve de Bas, Spain) following the manufacturer’s instructions with the following analysis kits: Total-cholesterol (1001095); HDL-cholesterol (1001096); LDL-cholesterol (41023); and triacylglycerides (1001312).

### 4.8. Western Blot Analysis

Total protein (15–40 μg) was resolved by SDS-PAGE and electroblotted onto polyvinylidene difluoride membranes (Immobilon-P Millipore, Bedford, MA, USA). Immunodetection was performed using as primary antibody porin (A31855, Molecular Probes, Eugene, OR, USA). A monoclonal antibody to β-actin (Sigma, USA) was used to control protein loading. Protein bands were visualized with the chemiluminescence ECL method (Millipore Corporation, Billerica, MA, USA). Luminescence was recorded and quantified in a Lumi-Imager equipment (Boehringer, Mannheim, Germany), using the Quantity One 4.6.5. software (Bio-Rad, Hercules, CA, USA).

### 4.9. Statistical Analysis

Data is presented as the mean ± standard deviation. Analysis of Gaussian distribution of all variables was performed by D’Agostino–Pearson omnibus normality test. The statistical significance comparison of variables with normal Gaussian distribution was calculated by one-way ANOVA. Multiple comparisons were performed comparing the mean of each group with the mean of the control normocaloric and correction for multiple comparisons was performed with Dunnett test. Non-normal Gaussian distribution was analysed by Kruskall–Wallis test with Dunn’s as a post-hoc test. In both cases, p-values below 0.05 in all statistical tests employed were considered as statistically significant. Partial least square-discriminant analysis (PLS-DA) was performed with transcriptomic and oxidative stress data to determine main differences between groups. Data was normalized by auto-scaling (mean-centred and divided by the standard deviation of each variable) for the PLS-DA analysis. Random forrest out-of-bag error (OOB) was performed to determine the probability that the variables employed in the PLS-DA analysis may discriminate the belonging of each animal to each group. Variable importance in projection (VIP scores) determined by the weight of the sum of squares of PLS loadings; was used to classify the main variables that contributed to the differences between groups. GraphPad Prism (version 5.0) and MetaboAnalyst (version 3.0) were used for statistical analysis and graph plotting.

## Figures and Tables

**Figure 1 ijms-20-05870-f001:**
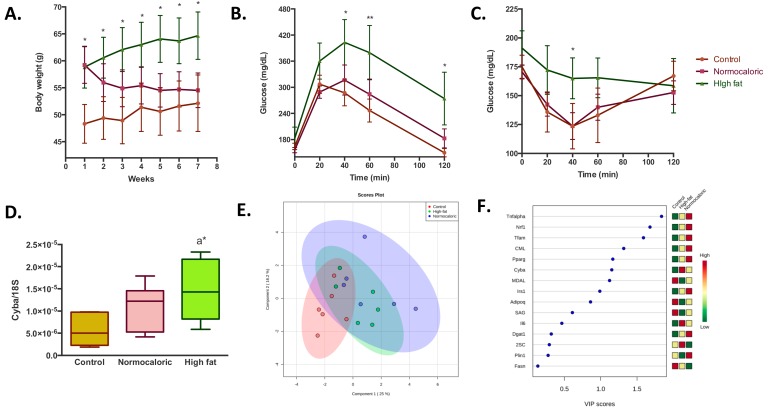
Adipose tissue metabolic changes induced by a high-fat diet and a weight reduction treatment through a normocaloric diet. Control: Mice fed with a normocaloric diet during the whole experimental period; high-fat: Mice fed with a high-fat diet during the whole experimental period; normocaloric: Mice fed with a high-fat diet during the first four months of experimentation and then fed with a normocaloric diet during the next two months as a treatment period. Data are presented as the mean ± standard deviation. (**A**) Body weight evolution during the treatment period. (**B**) and (**C**) glucose and insulin tolerance tests at the end of the experimental period. (**D**) Changes in the expression of *Cyba*. Data were normalized with endogenous control *18S* expression by 2^−∆Ct^. (**E**) Partial least-square discriminant analysis of transcriptomic and oxidative stress variables. The importance of each variable is represented in (**F**), squares on the right side of the graph represent the differences in the concentrations between the three groups. Red, yellow, and green squares indicate higher, intermediate, and lower concentrations, respectively of the variable in each group. * and ** denotes statistical difference with the normocaloric control group (a) with p-values below 0.05 and 0.01, respectively.

**Figure 2 ijms-20-05870-f002:**
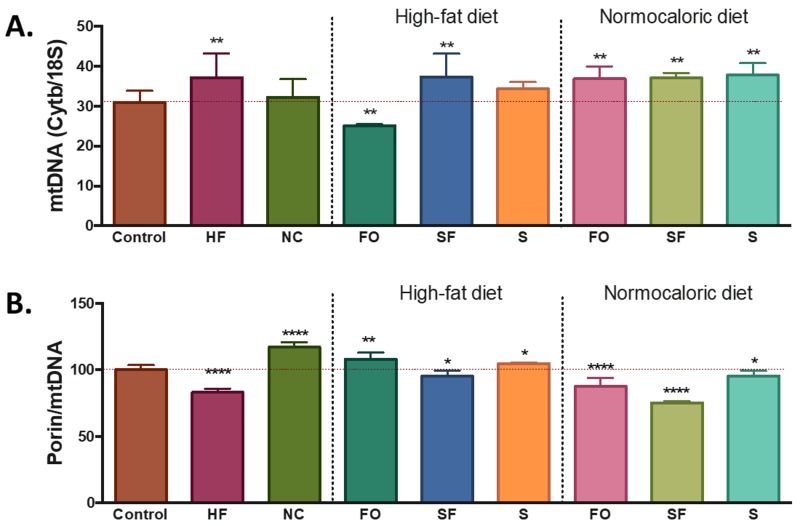
Changes in mtDNA and porin levels in adipose tissue. (**A**) mtDNA content, data was normalized with nuclear gene *18S* as the endogenous control. The expression was calculated by 2^−∆Ct^. (**B**) Porin content was determined by western blot. The levels of porin were normalized by the amount of mtDNA. HF = high-fat diet, NC = normocaloric diet, FO = fish oil, SF = soluble fibre, and S = soy. *, **, and **** denotes statistical difference with the control group with p-values below 0.05, 0,01, and 0.0001, respectively.

**Figure 3 ijms-20-05870-f003:**
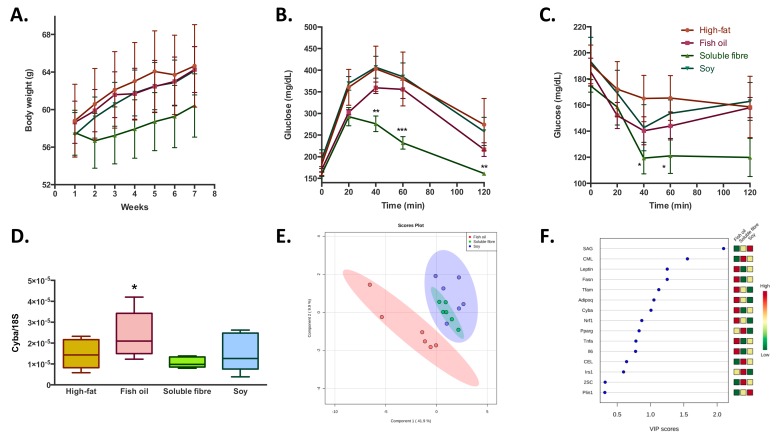
Adipose tissue metabolic changes induced by the treatment with fish oil, soluble fibre, and soy during the consumption of a high-fat diet. Data are presented as the mean ± standard deviation. (**A**) Body weight evolution during the treatment period. (**B**) and (**C**) glucose and insulin tolerance tests at the end of the experimental period. (**D**) Changes in the expression of *Cyba*. Data were normalized with endogenous control *18S* expression by 2^−∆Ct^. (**E**) Partial least square-discriminant analysis of transcriptomic and oxidative stress variables. The importance of each variable is represented in (**F**), squares on the right side of the graph represent the differences in the concentrations between the three groups. Red, yellow, and green squares indicate higher, intermediate, and lower concentrations, respectively of the variable in each group. *, ** and *** denotes statistical difference with the control high-fat group with p-values below 0.05, 0.01 and 0.001 respectively.

**Figure 4 ijms-20-05870-f004:**
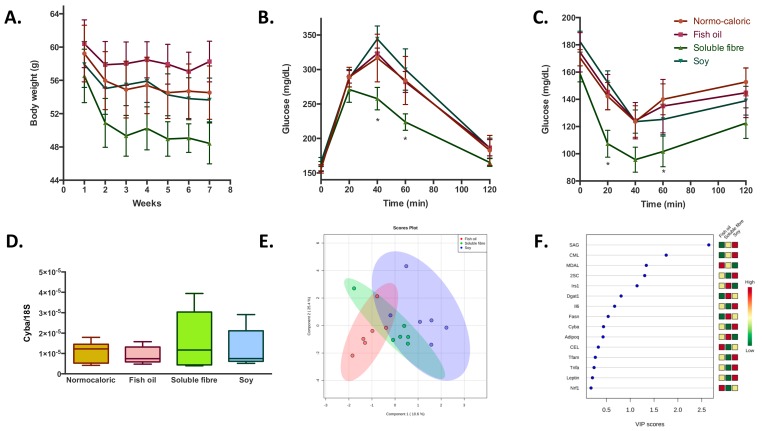
Adipose tissue metabolic changes induced by the treatment with fish oil, soluble fibre, and soy during the consumption of a normocaloric diet. Data are presented as the mean ± standard deviation. (**A**) Body weight evolution during the treatment period. (**B**) and (**C**) glucose and insulin tolerance tests at the end of the experimental period. (**D**) Changes in the expression of *Cyba*. Data were normalized with endogenous control *18S* expression by 2^−∆Ct^. (**E**) Partial least square- discriminant analysis of transcriptomic and oxidative stress variables. The importance of each variable is represented in (**F**), squares on the right side of the graph represent the differences in the concentrations between the three groups. Red, yellow, and green squares indicate higher, intermediate, and lower concentrations, respectively of the variable in each group. * denotes statistical difference with the control nomocaloric group with *p*-values below 0.05.

**Figure 5 ijms-20-05870-f005:**
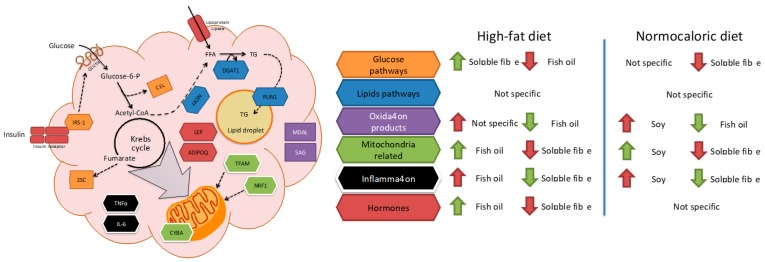
Schematic representation of the involvement of each analyzed bioactive food in adipose tissue metabolism. Green and red arrows represent the main effects of each bioactive food in the represented metabolic pathways. Green color identifies the bioactive food with a higher impact, while red color identifies the compound with minor effects in the pathways. A brief description of the pathways is the following: Insulin receptor substrate-1 (IRS-1) is phosphorylated after the stimulation of the insulin receptor and initiates insulin signaling which will induce glucose uptake by GLUT4 transporter (Glucose transporter). During the metabolization of glucose some by products may be produced such as carboxyethyl-lysine (CEL) (from methylglyoxal) and 2-succinyl-cysteine (2SC) (from fumarate). Acetyl-CoA could serve as an initial metabolite for the novo-lipogenesis, in which fatty acid synthase (FASN) and diacylglycerol acyltransferase-1 (DGAT1) are necessary for free fatty acid (FFA) and triacylglycerides (TG) synthesis, respectively, with its further incorporation into the lipid droplet and release for energy source is regulated by Perilipin-1 (PLIN1). Leptin (LEP) and adiponectin (ADIPOQ) are hormones secreted mostly in relation to the energy stored in the adipocyte and serve as signaling molecules for body’s energy homeostasis regulation. Mitochondrial Transcription Factor A (TFAM) and nuclear respiratory factor-1 (NRF1) are proteins implicated in mitochondrial DNA transcription and replication, which may modify the number and morphology of the mitochondria and its components such as cytochrome b-245 alpha (CYBA). Impaired mitochondrial activity may lead to the increase in oxidative stress compounds such as malondialdehyde-lysine (MDAL) and semialdehyde glutamic (SAG), which in some cases may induce inflammation with increased levels of tumor necrosis factor alpha (TNFα) and interleukin-6 (IL-6).

**Table 1 ijms-20-05870-t001:** Dietary and blood biochemical parameters.

Parameter	Control Normocaloric Diet	Control High-Fat Diet	Treatment Normocaloric Diet	Treatment High-Fat Diet + Fish Oil	Treatment High-Fat Diet + Soluble Fibre	Treatment High-Fat Diet + Soy	Treatment Normocaloric Diet + Fish Oil	Treatment Normocaloric Diet + Soluble Fibre	Treatment Normocaloric Diet + Soy
Diet intake (g/week/g body weight)	0.69 ± 0.12	0.47 ± 0.11^a****^	0.55 ± 0.06^a**^	0.41 ± 0.03 ^a****^	0.47 ± 0.05^a****^	0.44 ± 0.01^a****^	0.53 ± 0.03^a**^	0.55 ± 0.04^a**^	0.55 ± 0.05^a**^
Glucose (mg/dL)	172 ± 25	197 ± 62	155 ± 15	171 ± 16	160 ± 14	194 ± 53	157 ± 11	160 ± 17	166 ± 16
Cholesterol (mg/dL)	215 ± 61	227 ± 61	189 ± 55	208 ± 50	160 ± 23	195 ± 27	130 ± 30^a**^	135 ± 20^a*^	173 ± 30
HDL-cholesterol (mg/dL)	120 ± 22	117 ± 23	106 ± 24	117 ± 25	106 ± 13	139 ± 24	89 ± 9	100 ± 12	120 ± 7
LDL-cholesterol (mg/dL)	92 ± 25	104 ± 25	103 ± 45	98 ± 24	90 ± 9	120 ± 30	47 ± 14^a*,b**^	83 ± 22	103 ± 18
Triacylglycerides (mg/dL)	69 ± 23	63 ± 20	75 ± 17	70 ± 22	86 ± 8	69 ± 25	74 ± 28	68 ± 15	85 ± 27
Antioxidant capacity (FRAP) (µmol Trolox equivalents)	215 ± 65	322 ± 70	222 ± 76	272 ± 120	213 ± 39^c*^	238 ± 50	217 ± 59	240 ± 30	213 ± 59

Values are reported as the mean ± standard deviation. Differences between groups were determined by one-way ANOVA and multiple comparisons were performed comparing the mean of each group with the mean of the control normocaloric (a), control high-fat (b) and treatment normocaloric (c) diets. Correction for multiple comparisons was performed with Dunnett test and p-values below 0.05 (marked in bold) were considered as a significant difference between groups (* *p* < 0.05, ** *p* < 0.01, and **** *p* < 0.0001).

**Table 2 ijms-20-05870-t002:** Adipose tissue oxidative stress biomarkers.

Parameter	Control Normocaloric Diet	Control High-Fat Diet	Treatment Normocaloric Diet	Treatment High-Fat diet + Fish Oil	Treatment High-Fat Diet + Soluble Fibre	Treatment High-Fat Diet + Soy	Treatment Normocaloric Diet + Fish Oil	Treatment Normocaloric Diet + Soluble Fibre	Treatment Normocaloric Diet + Soy
Semialdehyde glutamic	7134 ± 222	5784 ± 1007	6745 ± 632	4642 ± 95^a**^	13406 ± 734^a****,c****^	14197 ± 2418^a****,c****^	5969 ± 271	14943 ± 1256^a****,b****^	12789 ± 1606^a****,b****^
Nε-(Carboxiethyl)lysine	345 ± 48	314 ± 57	342 ± 35	333 ± 68	419 ± 71	350 ± 86	383 ± 64	401 ± 45	363 ± 94
Malondialdehyde lysine	629 ± 104	566 ± 55	342 ± 101^a***^	275 ± 90^a****^	314 ± 113^a****^	258 ± 125^a****^	479 ± 164	535 ± 19	358 ± 108^a***,b*^
Nε-(Carboximethyl)lysine	1539 ± 122	1709 ± 323	1752 ± 137	1660 ± 140	2853 ± 329	2579 ± 361	1745 ± 138	2385 ± 334	2407 ± 562
2-succinyl-cysteine	43.6 ± 4.8	44.6 ± 7.2	42.3 ± 6.5	44.6 ± 8.9	46.4 ± 9.2	41.0 ± 1.4	42.3 ± 7.6	46.5 ± 2.1	42.5 ± 4.8

Values (µmol/mol of lysine) are reported as the mean ± standard deviation. Differences between groups were determined by one-way ANOVA and multiple comparisons were performed comparing the mean of each group with the mean of the control normocaloric (a), control high-fat (b) and treatment normocaloric (c) diets. Correction for multiple comparisons was performed with Dunnett test and p-values below 0.05 (marked in bold) were considered as a significant difference between groups (* *p* < 0.05, ** *p* < 0.01, *** *p* < 0.001, and **** *p* < 0.0001).

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
