# Peer review of "Adipose Tissue Mitochondrial Factors Profile after Dietary Bioactive Compound Weight Reduction Treatments in a Mice Obesity Model"

_ijms, 2019, doi:10.3390/ijms20235870_

Round 1

Reviewer 1 Report

This study has examined the ability of fish oil, soy and soluble fibre to decrease obesity and influence mitochondrial content and metabolism in adipose tissue in high fat fed mice.  Additionally the ability of these dietary compounds to improve glucose and insulin tolerance was examined.

Major comments

The study used n=6 mice in each group.  This is on the lower end of the number of mice usually included in animal studies.  Could the authors please comment on how they chose to use this many animals in each group?  It would be unfortunate if the study was underpowered as there is some results where there is a change, but it doesn't appear to be statistically significant (e.g. weight loss, blood glucose, gene expression).  Related to this there a few places in the discussion where changes in gene expression are mentioned, but these changes are not statistically significant (i.e. lines 272, 311, 324). 

Given the focus on adipose tissue metabolism it would have been interesting to see some further analysis on mitochondrial enzymes/proteins (enzyme activity, protein expression).  If it is not possible to include this in the current study it might be interesting to look at in the future.

Other comments

As glucose and insulin tolerance is measured in this study it might be good to include some information on this in the introduction and discussion. Line 61, perhaps be more specific on what exactly will be measured. Has plasma insulin been measured for fasting blood samples or during the glucose tolerance test?  It may aid interpretation of the GTT results. Line 67, weeks should be months? Line 115, please spell out MDAL. Line 117, please confirm the three groups for clarity. Figure 3 seems to be missing 'D', please correct on the figure and in the text. The second figure 3 should be figure 4, and the summary figure should be figure 5. Gene expression measurements, please confirm that 18S was a good choice (i.e. it was stable and didn't vary between groups) for this study.  Ideally more than one reference gene would be used.

Author Response

We want to thank the reviewer for the comments raised since it helped to improve the understanding and quality of the manuscript. In italic and bold is highlighted the detailed response to your comments.

This study has examined the ability of fish oil, soy and soluble fibre to decrease obesity and influence mitochondrial content and metabolism in adipose tissue in high fat fed mice.  Additionally the ability of these dietary compounds to improve glucose and insulin tolerance was examined.

Major comments

The study used n=6 mice in each group.  This is on the lower end of the number of mice usually included in animal studies.  Could the authors please comment on how they chose to use this many animals in each group?  It would be unfortunate if the study was underpowered as there is some results where there is a change, but it doesn't appear to be statistically significant (e.g. weight loss, blood glucose, gene expression). 

We apologize for not describing the rationale of the sample size selection. The sample size was estimated with an acceptable risk of 0.05 and an estimated power or 90%. It was considered that the treatment would be effective if it induces a reduction of 6.5 g of body weight in the following two months of treatment, and a standard deviation of body weight of 3.6 g was used for sample size calculation by the following formula

Sample size =2·(Za + Zb)2·s2/d2

Sample size= 2·(1.645+1.282)2·(3.6)2/(6.5)2 = 5.25

The standard deviation was described in the manuscript (line 69), which corresponds to mice fed with a normocaloric diet. We estimated that mice fed with a normocaloric diet would increase their body weight in 4.0 g in two months, which will be a final body weight of 52.3 g. Since overweight mice started the second experimental period with a mean body weight of 58.8, a good treatment response will be the reduction of the body weight to similar levels of a mice fed with a normocaloric diet (58,8-52,3= 6.5 g)

Related to this there a few places in the discussion where changes in gene expression are mentioned, but these changes are not statistically significant (i.e. lines 272, 311, 324). 

We have expressed these findings in the wrong way. The results expressed came from the PLS-DA analysis. We have correct this description error (lines 298, 338 and 349)

 Given the focus on adipose tissue metabolism it would have been interesting to see some further analysis on mitochondrial enzymes/proteins (enzyme activity, protein expression).  If it is not possible to include this in the current study it might be interesting to look at in the future.

Unfortunately, we are unable to present results related to mitochondrial enzyme activity, protein expression or even respiratory capacity. We have included these suggestions in line 378-380.

Other comments

As glucose and insulin tolerance is measured in this study it might be good to include some information on this in the introduction and discussion. Line 61, perhaps be more specific on what exactly will be measured.

The following phrase was included following your suggestion “in adipose tissue metabolic profile during weight reduction-treatments” (line 61)

Has plasma insulin been measured for fasting blood samples or during the glucose tolerance test?  It may aid interpretation of the GTT results.

Unfortunately, we have not collected blood samples for insulin determination during the subcutaneous glucose tolerance test. Glucose was determined with a portable glucometer with low quantities of blood. To determine blood’s insulin it will be necessary to sacrifice the mice since at least 50 microL of plasma (100 microL of blood) are needed for each point determination. For the 5 point curve, a total volume of 500 microL is necessary, which is around 17% of total blood volume in the mice. In general, without fluid replacement, approximately 10% of the total blood volume (0.75% of BW) can be safely removed at one time (https://www.jax.org/news-and-insights/2005/october/how-much-blood-can-i-take-from-a-mouse-without-endangering-its-health), therefore for its determination, it is necessary to increase the number of animals employed in the experimental procedure.

Line 67, weeks should be months?

We have change weeks for months.

Line 115, please spell out MDAL.

We have included MDAL description as Malondialdehyde-lysine

Line 117, please confirm the three groups for clarity.

The description of the three groups was included.

Figure 3 seems to be missing 'D', please correct on the figure and in the text. The second figure 3 should be figure 4, and the summary figure should be figure 5.

Figure numbers and descriptions were corrected.

Gene expression measurements, please confirm that 18S was a good choice (i.e. it was stable and didn't vary between groups) for this study.  Ideally more than one reference gene would be used

No differences in the expression of 18S was observed between groups as expressed in the material and methods section line 448.

Reviewer 2 Report

Healthy metabolism of adipose tissue is a key feature for the success of weight reduction in patients. The improvement of the metabolic capacity of adipose tissue of people with obesity is important.

The authors present herein a adipose tissue mitochondrial factors profile after dietary bioactive compound. The results clearly prove the soluble fibre supplementation, compared to an energy reduction program, was the only treatment able to induce a significant additional effect in the improvement of weight loss and adipose tissue metabolism.

The paper is well written, methodically sound and presents interesting results. I recommend acceptance of this manuscript with minor typing errors and: for example, you should change from μl to μL

You should checked the source (origin or company..) of apparatus and materials.

After minor revision as outlined above the paper should be accepted for publication.

Author Response

Healthy metabolism of adipose tissue is a key feature for the success of weight reduction in patients. The improvement of the metabolic capacity of adipose tissue of people with obesity is important.

The authors present herein a adipose tissue mitochondrial factors profile after dietary bioactive compound. The results clearly prove the soluble fibre supplementation, compared to an energy reduction program, was the only treatment able to induce a significant additional effect in the improvement of weight loss and adipose tissue metabolism.

The paper is well written, methodically sound and presents interesting results. I recommend acceptance of this manuscript with minor typing errors and: for example, you should change from μl to μL. You should checked the source (origin or company..) of apparatus and materials.

After minor revision as outlined above the paper should be accepted for publication.

We want to thank the reviewer for the positive assessment of the study. We have included the source of fish oil, soy and soluble fibre and checked the material methods sections for any missing data regarding materials and apparatus following your suggestion. The manuscript was reviewed again and some typing errors were corrected. The changes in the manuscript are highlighted in red by the track-changes option. 

Reviewer 3 Report

In this manuscript, the authors studied effects of fish oil, soluble fiber and soy modulating metabolic factors in adipose tissues using obesity mouse model induced by a high-fat diet. They concluded that fish oil improved mitochondrial related oxidation products in high fat diet mice, soluble fiber improved glucose and inflammation pathways. They also have summarized their findings nicely at the end of the paper, which is really helpful for the reader to connect all the dots.

Even though the authors have studied these in detail, this manuscript can not be accepted in its current form:

Following explanation or changes are warranted:

1) I would like the authors to explain that what are the implication of the studies? None of the treatment they had tried, have any effect on the total body weight gain of obese mice. (Figure-3A) I certainly see the effect on the mitochondrial factors and other factors but if the effect of that can not be translated in the system then they should have at least an explanation that "even if we do not observe changes in the body weight, but this imply that it creates some sort of vulnerability/improvement in the system that can be targeted by something else".

2) They should explain in the result sections some of their observation in detail. For example, line 169 to 179, they just mentioned all the enzymes levels in the text but did not imply where all these enzyme belong and how does it affect that particular pathway(s). They should elaborate in some detail.

3) Figure:1 A,B, and C has "control" but the figure and the explanation lacks the explanation what the control is. Please add details of the control group.

Also,  a study (https://www.ncbi.nlm.nih.gov/pmc/articles/PMC4882321/) was published couple of years ago, where a group showed that normocaloric diet can restore body weight in the obese mice. (As the title itself suggests) The authors should explain how contrary that is from their study and include argument in the discussion section.

Author Response

We want to thank the reviewer for the comments raised since it helped to improve the understanding and quality of the manuscript. In italic and bold are highlighted the detailed response to your comments.

In this manuscript, the authors studied effects of fish oil, soluble fiber and soy modulating metabolic factors in adipose tissues using obesity mouse model induced by a high-fat diet. They concluded that fish oil improved mitochondrial related oxidation products in high fat diet mice, soluble fiber improved glucose and inflammation pathways. They also have summarized their findings nicely at the end of the paper, which is really helpful for the reader to connect all the dots.

Even though the authors have studied these in detail, this manuscript can not be accepted in its current form:

Following explanation or changes are warranted:

1) I would like the authors to explain that what are the implication of the studies? None of the treatment they had tried, have any effect on the total body weight gain of obese mice. (Figure-3A) I certainly see the effect on the mitochondrial factors and other factors but if the effect of that can not be translated in the system then they should have at least an explanation that "even if we do not observe changes in the body weight, but this imply that it creates some sort of vulnerability/improvement in the system that can be targeted by something else".

We thank the reviewer for this comment since based on the suggestion we have improved the significance of the findings in the discussion section. We suggest that the lack of differences in metabolic biomarkers may be derived from differences in the respond of each organ to energy restriction. In this case, it seems that adipose tissue may be liable to return to a disease condition when there is an increase in energy intake. This observation may explain the deleterious effects of yo-yo dieting and why in some cases people with obesity are likely to increase their weight when dietary treatments are not followed after the weight reduction. This comment is included from line 382 to 390.

2) They should explain in the result sections some of their observation in detail. For example, line 169 to 179, they just mentioned all the enzymes levels in the text but did not imply where all these enzyme belong and how does it affect that particular pathway(s). They should elaborate in some detail.

A detailed explanation of the function and/or meaning of each analyzed metabolite or mRNA in adipose tissue metabolism are included in Figure 5. We appreciate this suggestion, since it may help the readers to understand the meaning of each modification induced by each dietary compound.

3) Figure:1 A,B, and C has "control" but the figure and the explanation lacks the explanation what the control is. Please add details of the control group.

A detailed explanation of each group in Figure 1 is included in the figure legend following your suggestion.

Also,  a study (https://www.ncbi.nlm.nih.gov/pmc/articles/PMC4882321/) was published couple of years ago, where a group showed that normocaloric diet can restore body weight in the obese mice. (As the title itself suggests) The authors should explain how contrary that is from their study and include argument in the discussion section.

We have included the findings of the suggested study in the discussion section line 383.

Round 2

Reviewer 1 Report

Thank you for addressing my previous concerns.

For future reference it is possible to measure plasma insulin on small volumes (5-10 μL in duplicate) using ELISA, and you can at the end of GTT give the mice an injection of saline to replace the blood volume lost.

Reviewer 3 Report

The authors have addressed all the concerns raised in my earlier review. Thus, this manuscript is now accepted in its current form.

One minor detail, figure:5 (Summary image) is listed as figure:4. (Line:277)